# Comparison of Safety and Efficacy of Hydrus and iStent Combined with Phacoemulsyfication in Open Angle Glaucoma Patients: 24-Month Follow-Up

**DOI:** 10.3390/ijerph20054152

**Published:** 2023-02-25

**Authors:** Joanna Jabłońska, Katarzyna Lewczuk, Marek Tadeusz Rękas

**Affiliations:** Department of Ophthalmology, Military Institute of Medicine—National Research Institute in Warsaw, 04-141 Warsaw, Poland

**Keywords:** iStent, Hydrus, glaucoma, microinvasive glaucoma surgery

## Abstract

The paper presents the results of a 24-month-long observation comparing the effectiveness and safety of two micro-invasive glaucoma surgery (MIGS) devices: Hydrus Microstent and iStent Trabecular Bypass in combination with cataract phacoemulsification in the treatment of open-angle glaucoma. We also analyzed the impact of preoperative factors on achieving surgical success in both surgical methods. The prospective, comparative, non-randomized study included 65 glaucoma surgeries. In 35 patients (53.8%), an iStent implant procedure was performed, while 30 patients (46.2%) underwent a Hydrus implant procedure. The demographic data was similar in both treatment groups. At 24 months after surgery, the mean IOP in the iStent group was 15.9 ± 3.0 mmHg and in the Hydrus group 16.2 ± 1.8 mmHg. The difference between the mean iStent vs Hydrus after two years of treatment was −0.3 (*p* = 0.683). At the 24 month follow-up, the average change in the number of antiglaucoma medications used was 71.7% in the iStent group and 79.6% in the Hydrus group. The difference in mean percentage change between groups was 7.9% in favor of Hydrus. Patients under the age of 70 may benefit from a greater risk reduction in the Hydrus group (HR = 0.81), while those over the age of 70 may benefit from a risk reduction in the iStent group (HR = 1.33). IOP cases with >18 mmHg before the surgery have a better chance of surgical success with the Hydrus method (HR = 0.28), and with IOP < 18 mmHg in the iStent group (HR = 1.93). Cases with more drugs (≥3 drugs) are more favorable in the Hydrus group (HR = 0.23), while those with a maximum of two drugs have a better prognosis in the iStent group (HR = 2.23). The most common postoperative complication was the presence of erythrocytes in the anterior chamber (AC), found in 40.0% of operated eyes in the Hydrus group. The profile of observed complications and significant improvement in visual acuity allows us to consider both implants as a safe way of treating patients with early or moderate glaucoma and co-existing cataracts.

## 1. Introduction

Glaucoma is a major public health problem and a leading cause of irreversible visual impairment worldwide. It is estimated that nearly 60 million people worldwide suffer from primary open-angle glaucoma (POAG). It is estimated that by 2040 this number will increase by 74% (111.8 million) [1,2]. The speed and risk of glaucoma development are affected by various progression factors. The main treatable risk factor for the development of glaucomatous neuropathy is elevated intraocular pressure (IOP), and its reduction to the target pressure remains the only proven method of preventing glaucoma damage. The first and most accessible therapy used in glaucoma is conservative treatment. If the target IOP cannot be reached and/or the disease progresses despite antiglaucoma medications, surgical IOP reduction is the next step [3]. Modern glaucoma surgery aims primarily to improve the patient’s safety and quality of life [4]. For over 10 years, we have been observing the intensive development of minimally invasive procedures and the changing divisions and definitions of this type of procedures. It is generally accepted that MIGS constitutes a diverse group of glaucoma surgeries intended to be safer and to cause much less tissue damage than traditional surgical procedures [5]. According to the position of the European Glaucoma Society (EGS) and the 2020 guidelines, the term MIGS covers only ab-interno procedures with efficacy independent of the filtration bleb [3]. According to the American Glaucoma Society and the US Food and Drug Administration (FDA), the term MIGS refers to the implantation of a surgical device intended to lower IOP via an outflow mechanism, with either an ab-interno or ab-externo approach, associated with very little or no scleral dissection.

iStent trabecular micro-bypass (Glaukos Corporation, Laguna Hills, CA, USA) is a one-piece heparin-coated titanium implant. It does not exhibit ferromagnetic properties, making it safe to use magnetic resonance imaging (MRI). A heparin coating is applied to the outer layer and the inner lumen. This helps to improve the flow of aqueous humor. The iStent is an L-shaped device. The length of the implant is 1 mm and the height is 0.33 mm. iStent was approved in 2012 by the FDA and is recommended in early and moderate glaucoma [5,6].

Hydrus (Ivantis, Inc., Irvine, CA, USA) is a flexible implant of the SC. It is made of nitinol, a biocompatible material that has shape memory properties and is widely used in vascular surgery. The Hydrus microstent is an 8 mm long crescent-shaped implant with an open structure, adapted to the course of the SC. It opens towards the back and has 3 windows along its course. After implantation, the microstent bypasses the trabeculum and dilates the SC over 3–4 clock hours (one quadrant), providing a route for outflow to multiple collector channels. The increase in the transverse dimension of the SC after implantation is 241 µm, which is about 4–5 times more than the natural cross-section of the canal. The Hydrus implant was approved by the FDA in August 2018 [6,7].

The aim of this study is to determine and compare the IOP-lowering potential of two modern iStent and Hydrus implants, which are permanently included in the surgical treatment of glaucoma, to assess the stability of the effect obtained after their implantation, as well as to determine the safety profile.

## 2. Materials and Methods

A prospective non-randomized comparative study included 65 glaucoma surgeries. 35 patients (53.8%) had surgical treatment with the 1st generation iStent implant, while 30 patients (46.2%) underwent surgery with the Hydrus implant. All surgeries were combined with the phacoemulsification cataract procedure. The study was conducted in the Department of Ophthalmology at Military Institute of Medicine—National Research Institute in Warsaw, Poland. The study followed the tenets of the Declaration of Helsinki, and was approved by the Bioethics Committee at the Military Institute of Medicine (75/WIM/2015). All participants gave informed written consent. Inclusion criteria included cataracts that had a significant impact on the patients’ visual acuity and one of the following conditions: progression of primary open-angle glaucoma (POAG) confirmed by two consecutive tests with the Humphrey Visual Field Analyzer (Carl Zeiss AG, Germany) using the SITA Standard 24-2 algorithm despite antiglaucoma medications (1 to 4 active ingredients) or intolerance or non-compliance with treatment. The exclusion criteria were: lack of consent for study participation, narrow-angle or closed-angle glaucoma, secondary glaucoma (including pseudoexfoliation syndrome, pigmentary glaucoma), juvenile glaucoma, advanced glaucoma, severe proliferative diabetic retinopathy, cloudy cornea, advanced macular degeneration, history of antiglaucoma surgical interventions (as well as laser trabeculoplasty), and use of more than four antiglaucoma medications.

### 2.1. Preoperative and Postoperative Examination

Results were collected during follow-up visits performed as follows: at 1 day, 7 days (±2 day), 1 month (±7 days), 3 months (±14 days), 6 months (±21 days), 12 months (±30 days), 18 months (±42 days), 24 months (±60 days), and 30 months (±60 days). Preoperative examination included obtaining patient data (age, sex, antiglaucoma medications, and surgery procedures). Basic procedures included IOP measurement using a Goldman applanation tonometer, best-corrected visual acuity (BCVA) using the Snellen chart, anterior segment assessment with funduscopic examination using a slit lamp, and Goldmann three-mirror lens for gonioscopy assessed with Schaffer classification. Postoperatively, BCVA, IOP, and number of antiglaucoma medications were assessed, and the anterior segment and fundus were examined at each control visit. Postoperative complications included hypotony (defined as an IOP < 6 mmHg), microhyphema, keratitis, peripheral anterior synechiae (PAS) (Figure 1), device migration or dislocation. All antiglaucoma medications were withdrawn after the surgery. When the target IOP was not achieved after surgery it was re-administrated according to the EGS guidelines [3]. After surgery, all patients were prescribed antibiotic and anti-inflammatory eye drops for 4 weeks.

### 2.2. Surgical Technique

All surgical procedures were performed under local anesthesia by one surgeon. Cataract phacoemulsification with implantation of an artificial posterior-chamber lens insertion into the capsular bag was first performed in all patients, and subsequently, the implantation of the Hydrus microstent or iStent in the SC was undertaken. Using a direct gonioscope lens (Swan-Jacobs gonioscope), an additional amount of viscoelastic substance was administrated into the anterior chamber. The injector ended with a cannula with a Hydrus micro-implant, or a 23 gauge applicator was introduced through the same temporal incision that was used during the cataract stage. The microstent was then advanced to span approximately 90 degrees of SC, while the 1–2 mm inlet segment was left to reside in the AC. For iStent implantation, a 23 gauge applicator was used. iStent was gently advanced through the trabecular meshwork and placed in the SC in the inferior nasal position (3:00 to 4:00 h in the right eye, 8:00 to 9:00 h in the left eye). After confirming the proper placement of the device, the implants injector was withdrawn and the viscoelastic was removed.

### 2.3. Surgical Success

The effectiveness of the procedures used in the study was analyzed on the basis of all available data from IOP measurements and the number of antiglaucoma medications. Surgical success was defined as freedom from secondary surgery, an IOP of 18 mmHg or less, and discontinuation of all antiglaucoma medications. The primary parameters were the assessment of IOP reduction for each eye at the time of observation, and the reduction of the number of antiglaucoma medications used in relation to preoperative values.

Two criteria were adopted:IOP ≤ 18 mmHg;IOP ≤ 15 mmHg.

Complete success was defined as the percentage of cases meeting the IOP criteria without the use of antiglaucoma medications, while satisfactory success was defined as the percentage of eyes meeting the IOP criteria with additional antiglaucoma medications.

### 2.4. Statistical Evaluation

Statistical analysis was carried out according to the World Glaucoma Association Guidelines on Design and Reporting of Glaucoma Surgical Trials. All analyses were performed using the Statistical Analysis System (SAS) statistical software package v. 9.1.3. (SAS Institute Inc., Cary, NC, USA). Statistical analyses of the tested parameters for a given variable distribution were performed based on parametric and non-parametric tests. Knowledge of the type of feature distribution was determined on the basis of the Shapiro–Wilk test. To compare the significance of differences for two dependent samples, the *t*-Student test, the Wilcoxon test of ranks of pairs, was used, assuming a critical significance level of *p* < 0.05. The significance of differences between the means for the Hydrus vs. iStent method, in the case of features meeting the assumptions of a normal distribution, was verified with the Student’s *t*-test for independent observations. If the condition of normality was not met, the Mann–Whitney U test was used. The Fisher test was used to verify the difference in the proportions of subgroups in both treatment groups. Survival analysis was performed using the Kaplan–Meier method, and the significance between curves was determined using the log-rank test. The analysis of the risk of surgical failure between the Hydrus and iStent methods was performed using the Cox proportional hazards model.

## 3. Results

Sixty five eyes of 65 subjects were included in the study. The demographic data and baseline characteristics are presented in Table 1.

### 3.1. Intraocular Pressure

The comparison of the mean IOP values before surgery versus the values 12 and 24 months after surgery for the iStent and Hydrus implants is presented in Table 2.

In addition to the analysis of mean IOP before and after the procedure, the significance of differences between the methods in particular periods after surgery was verified. Except for the first days after surgery, the t-Student test for independent observations showed no statistically significant differences in mean IOP values between the iStent and the Hydrus implant. A summary of the IOP values for the iStent and Hydrus methods in particular postoperative periods is presented in Figure 2.

### 3.2. Medication

In the 24th month of observation, the mean change in the number of antiglaucoma medication used in the iStent group was 71.7%, while in the Hydrus group it was 79.6%. The difference in mean percent change between groups is 7.9% in favor of Hydrus (*p* = 0.666) (Table 3).

Figure 3 shows the average percentage changes with 95% confidence intervals for the iStent and Hydrus groups at each time point. In addition, the bubble chart shows the number of antiglaucoma medications used in particular periods after surgery.

### 3.3. Surgical Success

In the group of patients with the Hydrus implant, complete surgical success for the criterion IOP ≤ 18 mmHg at 12 months was achieved in 78.4% of cases, while in the iStent group the result is lower and amounts to 71.0%. A slightly opposite trend was obtained after 24 months. Here, in the Hydrus group, full surgical success was achieved for IOP ≤ 18 mmHg in 53.7% of cases, and in the iStent group—in 63.9% of cases. Complete success curves for IOP ≤ 18 mmHg indicate that by Month 24, the Hydrus Implant has a higher proportion of subjects with a stable IOP below 18 mmHg who did not require antiglaucoma medication after surgery compared to the iStent. However, since the 24th month, the trend is changing in favor of the iStent group. Nevertheless, it is difficult to talk about the statistical significance of this relationship. Comparing the survival distributions for the Hydrus vs. iStent group using the log-rank test, we obtain a *p*-value of 0.950 (Figure 4).

For the IOP criterion ≤ 15 mmHg, complete surgical success was achieved after 12 months in 43.1% of cases in the Hydrus group and 44.6% of cases in the iStent group, and after 24 months—in 23.5% of cases in the Hydrus group and 40.1% from the iStent group. Analyzing the course of survival curves for the IOP ≤ 15 mmHg, we see that in the first months after surgery, we have a higher percentage of patients for the Hydrus method than the iStent method with an IOP not exceeding 15 mmHg, without the need for antiglaucoma medications. Nevertheless, with the passage of time after surgery, this trend changes in favor of the iStent method (Figure 5).

Figure 6 shows hazard ratio estimates for Hydrus and iStent at 24 months postoperatively.

After analyzing HR, the results showed that gender, the presence of reflux before surgery, gonioscopy, glaucoma stage, central corneal thickness and degree of opacity of the lens had no effect on reducing the occurrence of an adverse event between the groups. HR in these cases is close to 1. the patient’s age, if below 70 years of age, may have a more favorable effect on risk reduction in the Hydrus group than in the iStent group (HR = 0.81), while the age of patients over 70 years of age may reduce the risk in the iStent group (HR = 1.33). The number of patients aged 70 years and older was 24 in the iStent group and 21 in the Hydrus group. Patients with a higher IOP before surgery have a better chance of surgical success with the Hydrus method (HR = 0.28) than with the iStent method. For patients with IOP < 18 mmHg, the prognosis is better in the iStent group (HR = 1.93). Patients with more antiglaucoma medications (≥3 drugs) fare better in the Hydrus group than in the iStent group (HR = 0.23). On the other hand, patients on either one or a maximum of two antiglaucoma medications have better prognosis in the iStent group (HR = 2.23 for two drugs, HR = 1.58 for one drug).

### 3.4. Safety

Safety included the BCVA and intraoperative and postoperative complications. The best corrected visual acuity before surgery was at a comparable level for both methods and was 0.3 ± 0.21 for the iStent method and 0.3 ± 0.51 for the Hydrus method. After 24 months from the procedure, the mean BCVA value for the iStent method was 0.1 ± 0.11, and for Hydrus—0.1 ± 0.12. No significant difference in mean BCVA was observed between the methods. No intraoperative complications were observed either in the iStent group or in the Hydrus group. The most common postoperative complication was the presence of erythrocytes in the AC. The number of cases observed was 12 (18.5% of the total), all of which belong to the Hydrus group (40.0% of operated eyes in this group). All these cases spontaneously resolved within a week, without any additional treatment. Six cases of increase in IOP ≥ 10 mmHg from baseline (9.2% of the total) were reported, of which two cases were in the Hydrus group (6.7% of operated eyes in this group), and four cases in the iStent group (11.4% operated eyes in this group). In the Hydrus group there were six cases of PAS (representing 20.0% of operated eyes in this group) and three cases of corneal oedema (representing 10.0% of operated eyes in this group). In one case in the Hydrus group, hypotony occurred on the 7th day after surgery and spontaneously resolved within 1 week without the need for additional interventions. Table 4 contains all complications.

## 4. Discussion

The iStent and Hydrus implants are used during cataract surgery to improve the conventional aqueous outflow. They belong to the group of microinvasive procedures that combine ab-interno access, minimal invasiveness for tissues, effectiveness of procedure and high safety. So far, there have been significantly more clinical papers on the iStent implant [8,9,10,11,12,13,14,15,16,17,18] than publications on the Hydrus implant in the surgical treatment of glaucoma [17,19,20,21,22,23,24]. Few publications compare both implants: The COMPARE study compares the Hydrus implant with two iStent implants [20], Holmes compares the Hydrus implant with the iStent inject implant [21], Lee describes the postoperative outcomes of phacoemulsification alone compared to combined phacoemulsification and iStent or Hydrus for open angle glaucoma [22]. This paper presents a comparison of their efficacy and safety in combined surgery with phacoemulsification of cataracts in patients with open-angle glaucoma, additionally extended by analysis of the impact of preoperative factors on achieving surgical success in both surgical methods. Taking into account the lack of randomization and washout, the study should be considered as burdened with some error. The above limitations are common in many publications, and the standardization of MIGS testing is still being refined. Therefore, the comparison of our results with the works of other authors is difficult due to the different methodological structure, the size of the study groups, the time of observation and different success criteria.

In this study, we assumed an IOP of 18 mmHg as a condition for success, similar to the authors of the COMPARE study. In our analysis, we adopted an additional IOP criterion of 15 mmHg. Due to the lack of a preoperative washout and the low preoperative mean IOP of 16.1 ± 3.2 mmHg in the iStent group and 16.3 ± 2.2 mmHg in the Hydrus group, the assumption of surgical success at the level of 21 mmHg, as many other authors have stated, would mean the work would be considered highly effective and the results could be unreliable. In addition, these are IOP values that are more clinician-acceptable when treating patients with moderate glaucoma. The adoption of such criteria is based on the Advanced Glaucoma Intervention Study (AGIS), which showed that patients with the most stable visual fields have an IOP of less than 18 mmHg [25]. In the two-year follow-up, both implants achieved similar indices in IOP regulation, and the difference in the obtained results was not statistically significant in individual treatment groups (*p* > 0.05). Clinical studies currently available in the literature show similar efficacy in reducing the IOP of each implant [12,14,19]. The most extensive study evaluating the effectiveness of the Hydrus implant is the prospective, multicentered, randomized Horizon Study [17]. The study compares the results of combined cataract surgery with Hydrus to cataract surgery. In the 24-month observation of 369 eyes in the Hydrus group, a 20% reduction in IOP was obtained in 77.3% of patients. In the Pfeiffer study of 100 eyes, the same reduction was achieved in 80% of cases [26]. In our work, although in the Hydrus group we did not observe significant reductions in the average IOP values (reduction by 0.09, on average 0.40%), we obtained a significant reduction in the number of antiglaucoma medication. After 24 months, the reduction was 79.55%. This is consistent with the observations of other authors [17,26]. The studies conducted by Samuelson and Craven for the iStent Study Group are the most extensive randomized studies of the iStent implant to date [9,12]. A 20% reduction in IOP was achieved in 66% of cases after 24 months of follow-up [12]. In our study, for the iStent method, an IOP reduction of at least 20% occurred in only 23.3% of cases over the same time period. It should be noted that a washout was performed in the study by Samuelson and Craven, and that the study included cases of pigmentary glaucoma and pseudoexfoliation syndrome (PEX) in addition to POAG, therefore a direct comparison with our results is not objective.

Most studies evaluating implant performance are based on a procedure combined with cataract phacoemulsification [9,12,18,19]. Cataract surgery alone can reduce IOP in the range of 1 to 4 mmHg [27]. All studies conducted comparing cataract surgery alone to combined surgery with Hydrus or iStent implantation have shown that the combination of these procedures has the added benefit of a greater IOP reduction, and in the number of antiglaucoma medications required [19]. In this study, we did not obtain a significant reduction in IOP in any of the groups. Most patients achieved IOP reduction by at least 10%, in the iStent group—46.6% of cases, and in the Hydrus group—47.8% of cases. In most published studies, the average IOP reduction for MIGS procedures combined with cataract phacoemulsification ranges from 12.7% to 20% with the Hydrus implant [15,16,17,20], and from 1.5% to 24% for s single iStent implantation [9,12,13]. In our study, in the case of both implants—iStent and Hydrus—the optimal hypotensive effect with the lowest possible number of antiglaucoma medications was obtained within the time range of 3 to 6 months after the operation. The Manchester iStent Study shows a similar relationship [13]. Samuelson [9] and Craven [12] report similar observations in single iStent implantation studies, reporting optimal IOP values from 3 to 6 months of follow-up. In the case of the Hydrus implant in the Horizon Study, the hypotensive effect is also delayed and has a prolonged effect up to the 24th month [17]. This delay and its stabilization over time can be attributed to the widening of the SC, followed by a gradual, progressive one during cannulation of the implanted stent.

Two recent retrospective studies compared the outcomes of the Hydrus and the iStent combined with phacoemulsification. For the iStent group, Lee included patients implanted with one first-generation iStent [22], and Holmes included patients implanted with a second-generation iStent inject [21]. For medication use, Lee reported a 0.5 medication reduction advantage with the Hydrus group compared with the iStent group [22], while Holmes reported an additional reduction with the iStent inject group by 0.5 medication on average compared with the Hydrus group [21].

In this paper, although it is difficult to talk about statistical significance in the comparison with the achieved surgical success between implants (*p*-value of 0.95), there is a certain trend. Regardless of the success criterion, in the first year after surgery, the use of the Hydrus implant reduces the risk of IOP increase compared to the iStent implant. In the case of complete surgical success for the IOP criterion of ≤18 mmHg, the risk of IOP increase is reduced by 46.1% in the Hydrus group compared to the iStent group. Taking into account the possibility of using at most one antiglaucoma medication, the risk reduction is even greater in favor of the Hydrus method and amounts to 50.3%. The risk reduction in favor of the Hydrus method is also noticeable in the case of the IOP criterion of ≤15 mmHg, however, on a much smaller scale. When examining the influence of preoperative factors that may differentiate the effectiveness of both implants, certain relationships emerge. Namely, factors such as gender, angle width, glaucoma severity, CCT and stage of cataract did not determine differences in achieving surgical success between the groups. The age of the patient below 70 years of age may have a more favorable effect on risk reduction in the Hydrus group than in the iStent group (HR = 0.81), while the age above 70 years of age may reduce the risk in the iStent group (HR = 1.33). We also analyzed the presence of blood reflux in the SC during preoperative provocative gonioscopy. We did not note the impact of this factor on the difference in achieving operational success between individual groups. An important element is the height of the preoperative IOP. The effect of this factor on postoperative IOP reduction is reported by most authors in studies of both iStent and Hydrus [9,12,16,17]. In our study, IOP > 18 mmHg before surgery increased the chances of surgical success in the Hydrus group (HR = 0.28), while for patients with IOP < 18 mmHg, the prognosis was better in the iStent group (HR = 1.93). In addition to IOP, the second factor that clearly differentiates individual methods is the number of antiglaucoma medication used before surgery. Cases with more medication (≥3 medications) fare better in the Hydrus group than in the iStent group (HR = 0.23). On the other hand, subjects on one or a maximum of two antiglaucoma medications have better prognosis in the iStent group (HR = 2.23 for 2 medications, HR = 1.58 for 1 medication). Better results in the Hydrus group with higher baseline IOP and more antiglaucoma medication taken pre-operatively support the hypothesis that Hydrus, due to its size and potential access to more collector channels, may regulate the flow of aqueous humor more effectively than with the iStent [28]. In addition, as in canaloplasty, a stretching effect of the trabecular meshwork may occur after implantation of the Hydrus implant. Lewis achieved a 40% reduction in IOP in eyes in canaloplasty, using ultrabiomicroscopic examination and trabecular stretching. Meanwhile in the eyes in which this stretching did not occur or only to a minimal extent, the reduction was only 24% [29]. Trabeculum stretching is one of the effects of pilocarpine. It is possible that implantation of the Hydrus implant may have a similar effect.

Most of the available publications document the improvement of BCVA after combined surgery with Hydrus or iStent implants [9,12,24], and are consistent with our results. We did not report any intraoperative complications in any of the study groups. In the literature, these complications are rare, and the most common include bleeding that prevents stent implantation and displacement of the implant [17,24]. The most common postoperative complication observed in our study was the occurrence of erythrocytes in AC. This event occurred only in the Hydrus group, in a total of 12 cases (40%). This complication concerned the immediate postoperative period and resolved after 48 h. In no case did we observe hyphema (blood level) in the AC. The literature does not mention the presence of erythrocytes in the AC, but the transient hyphema or microhyphema in the AC is described. The comparison of this complication between different studies is burdened with a certain error, resulting from different definitions and classifications. This event is more common with the Hydrus implant (0.5% to 19%) [16,17] than with the iStent implant (4%) [19]. The presence of erythrocytes or hyphema in the AC is associated with reflux of blood from the SC to the anterior chamber during stent implantation and is a common intraoperative process. It can be seen as a positive and normal symptom that occurs when the implant is properly positioned in the SC. The higher incidence of this complication in the Hydrus group is most likely related to the larger implant size.

Significant observation due to the frequency of occurrence was a transient increase in IOP ≥ 10 mmHg from baseline. This complication was noted in 6.7% of cases in the Hydrus group and in 11.4% in the iStent group. The odds of this happening in our study were 1.8 times greater for the iStent group than for the Hydrus group. In the literature, IOP increases are also higher for the iStent implant (from 1.1% to 21%) than for the Hydrus implant (from 4.76% to 6.5%) [14,19]. In our observations, the adverse event characteristic of the Hydrus group was the formation of PAS. We noted PAS in 20% of operated eyes in this group. The developed peripheral synechiae occurred as focal adhesion of the iris to the implant and were located near its inlet (Figure 1). The presence of PAS did not cause implant occlusion in any of the cases. In the works of other authors, PAS appears equally often in 14.9% of the Horizon Study [17], in 19% of Pfeiffer’s [26] and in 19% of Gandolfi [16]. In cases where the PAS is located at the inlet of the stent, obstruction is possible, as noted in a small percentage of cases by Pfeiffer and Samuelson in the Horizon Study [17,26]. Iris adherence followed by implant obstruction is reported in some iStent observations (3–4.3%) [9,12], but is much less frequent than PAS in the Hydrus implant, which may be related to the size of the stent. The small size of the iStent may also explain the cases of migration (2.6%) [12]. Despite the small size of the study groups, we observed hypotony, which is rare in MIGS procedures. This was in one case in the Hydrus group. Hypotony developed on the 7th day after surgery and spontaneously resolved within 1 week without the need for additional interventions. Transient hypotony was described by Samuelson in the case of the iStent [9]. As in our observation, this case did not require additional therapeutic measures. Risk of postoperative hypotension for SC-implanted stents is low because the lowest IOP achievable by bypassing the trabeculum is a few mmHg higher than the EVP.

The study has several significant limitations, as the lack of preoperative washout and randomization does not allow for a direct comparison or reliable assessment of the effectiveness of both procedures. It seems that the greatest benefits from the use of implants are visible in the reduction of the number of antiglaucoma medication. This observation may be very important, especially in the case of patients not tolerating or adhering to the treatment. Despite these limitations, we believe that this study is relevant in providing more evidence of the safety and efficacy of the Hydrus and iStent implants in clinical practice. Further studies, particularly randomized controlled trials, comparing two devices with different antiglaucoma surgeries, are needed with a longer follow-up.

## 5. Conclusions

The iStent and Hydrus implants used during cataract surgery to improve the conventional aqueous outflow to a similar extent allow for lowering of the intraocular pressure in the 24-month observation. In the absence of normalization of intraocular pressure after surgery, the number of antiglaucoma medications used to support pressure regulation is lower than before the operation. The degree of reduction in the number of medications used and the risk of increased intraocular pressure in favor of the Hydrus method are noticeable during the observation. Factors that affect post-operative success include age, intraocular pressure level and the number of antiglaucoma medications used before surgery. The profile of observed complications and significant improvement in visual acuity allows for the consideration of both implants as a safe way of treating patients with early or moderate glaucoma and co-existing cataracts.

## Figures and Tables

**Figure 1 ijerph-20-04152-f001:**
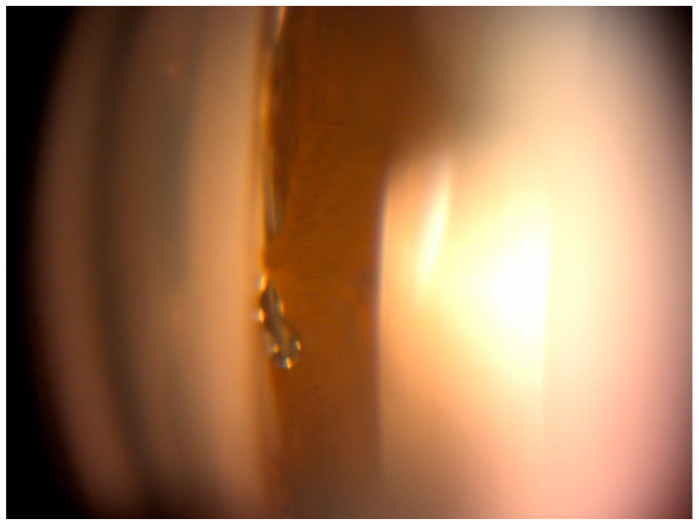
Gonioscopic view of Hydrus microstent showing peripheral anterior synechiae.

**Figure 2 ijerph-20-04152-f002:**
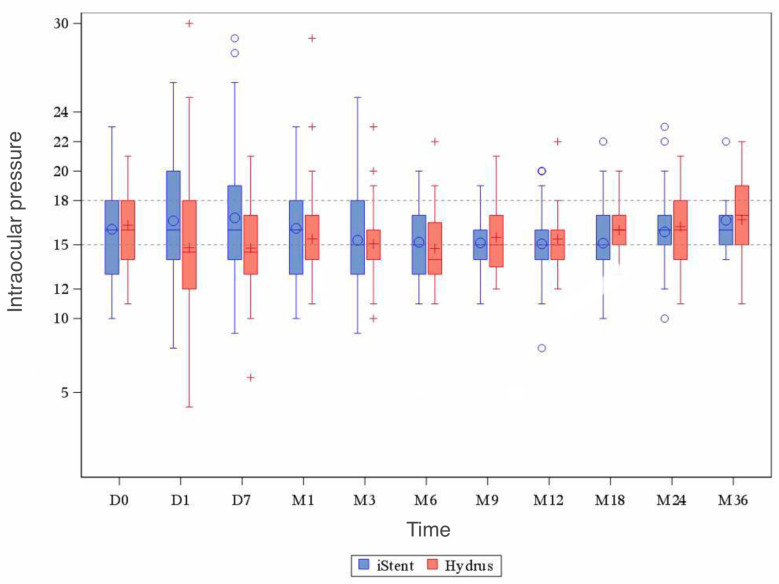
IOP values for the iStent and Hydrus methods in individual periods after surgery.

**Figure 3 ijerph-20-04152-f003:**
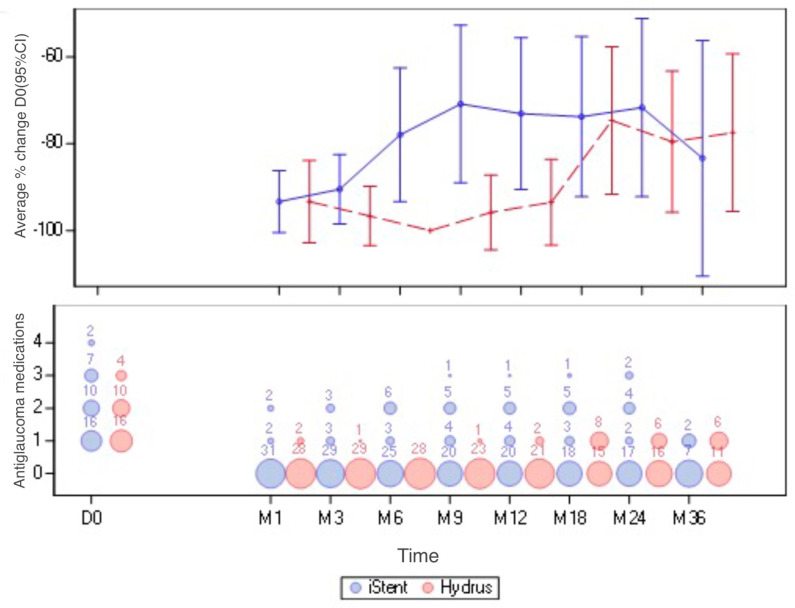
Mean percentage changes with 95% confidence intervals for iStent and Hydrus and the bubble chart with the number of antiglaucoma medications at each time point.

**Figure 4 ijerph-20-04152-f004:**
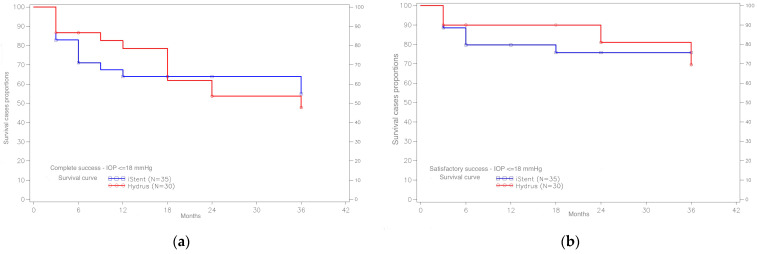
(**a**) Distribution of survival curves for complete surgical success for the IOP criterion ≤ 18 mmHg, for the Hydrus and iStent, and (**b**) distribution of survival curves for satisfactory surgical success for the IOP criterion ≤ 18 mmHg, for the Hydrus and the iStent.

**Figure 5 ijerph-20-04152-f005:**
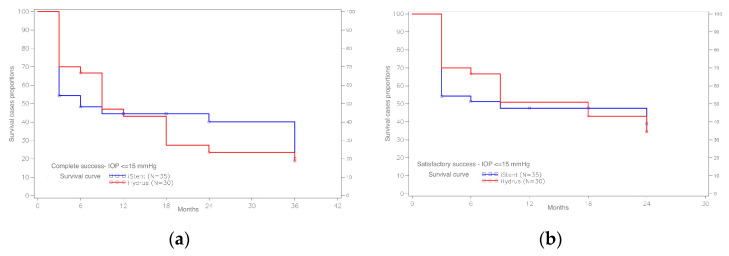
(**a**) Distribution of survival curves for complete surgical success for the IOP criterion ≤ 15 mmHg, for the Hydrus and iStent, and (**b**) distribution of survival curves for satisfactory surgical success for the IOP criterion ≤ 15 mmHg, for the Hydrus and the iStent.

**Figure 6 ijerph-20-04152-f006:**
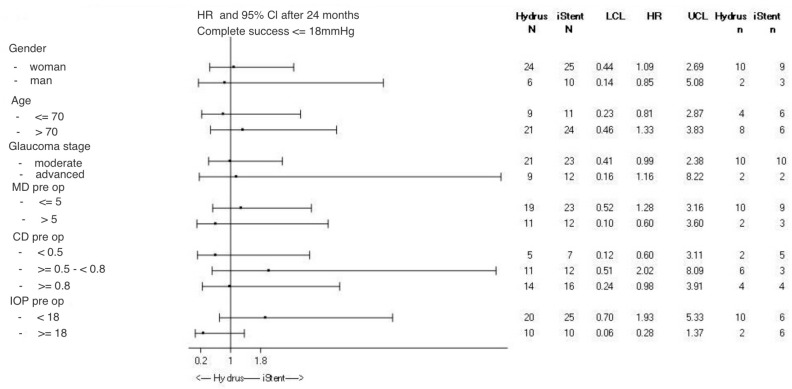
Forest plot for estimation of hazard ratio for Hydrus and iStent at 24 months after surgery.

**Table 1 ijerph-20-04152-t001:** Demographic and baseline characteristics of eyes meeting the criteria.

Parameter	iStent n 35	Hydrus n = 30	*p*-Value
Eyes, % right	16 (45.7)	17 (56.7)	0.379 ^a^
Gender, % female	25 (71)	24 (80)	0.424 ^a^
Age, mean (SD)	72.5 (7.5)	73.5 (11.4)	0.274 ^b^
Follow-up period (months) (SD)	23.06 (10.2)	25.90 (12.6)	0.177 ^b^
BCVA, mean (SD)	0.3 (0.21)	0.3 (0.15)	0.794 ^b^
IOP, mean (SD)	16.0 (3.2)	16.3 (2.2)	0.650 ^b^
CCT, mean (SD)	530 (28)	536 (43)	0.727 ^b^
Antiglaucoma medications %			
1	45.7 (16)	53.3 (16)	
2	28.6 (10)	33.3 (10)	
≥3	25.7 (9)	13.3 (4)	

^a^ Chi-square test ^b^ Wilcoxon’s test.

**Table 2 ijerph-20-04152-t002:** Comparison of mean IOP values before surgery vs. values 12 and 24 months after surgery.

**iStent**	** *n* **	**Mean**	**95% CI**	***p*-Value**
D0	35	16.1	(14.96–17.16)	
M12	30	15.1	(14.03–16.10)	
(D0-M12)	30	1.4	(0.38–2.34)	0.008
M24	25	15.9	(14.64–17.12)	
(D0-M24)	25	0.9	(−0.34–2.10)	0.151
**Hydrus**	** *n* **	**Mean**	**95% CI**	***p*-Value**
D0	30	16.3	(15.48–17.18)	
M12	23	15.4	(14.47–16.31)	
(D0-M12)	23	0.7	(−0.63–2.02)	0.288
M24	22	16.2	(14.99–17.47)	
(D0-M24)	22	0.1	(−1.26–1.45)	0.891

**Table 3 ijerph-20-04152-t003:** Percentage reduction in the number of antiglaucoma medications before surgery, and 12 and 24 months after surgery.

	*n*	Average Percentage Reduction	SD
**12 months after surgery**			
iStent	30	−73.1	46.72
Hydrus	23	−93.8	22.42
Mann–Whitney U Test	*p*-value = 0.038		
**24 months after surgery**			
iStent	25	−71.7	49.77
Hydrus	22	−79.6	36.71
Mann–Whitney U Test	*p*-value = 0.038		

**Table 4 ijerph-20-04152-t004:** List of postoperative complications.

Adverse Event	Hydrus N = 30	iStent N = 35	95% CI
Blood cells in AC	12 (40.0)	0	
Increase in IOP ≥ 10 mmHg	2 (6.7)	4 (11.4)	1.8 (0.3–10.6)
PAS	6 (20.0)	0	
Hypotony (IOP < 6 mmHg)	1 (3.3)	0	
Corneal oedema	3 (10.0)	0	

## Data Availability

All materials and information are available upon e-mail request to the corresponding author. The names and exact data of the study participants may not be available because of privacy policies.

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
