# Peer review of "Comparison of Safety and Efficacy of Hydrus and iStent Combined with Phacoemulsyfication in Open Angle Glaucoma Patients: 24-Month Follow-Up"

_ijerph, 2023, doi:10.3390/ijerph20054152_

Round 1
Reviewer 1 Report
I would like to point to the authors one minor inconsistency in the
section 2.3. about Surgical Success
Local antihypertensive drugs have been mentioned as the first. However, few sentences later there are hypotensive medication mentioned, then later antiglaucoma medication. Similar terms are in following parts of manuscript (antiglaucoma/anti-glaucoma, drugs/medication). I believe this inconsistency should be addressed before publishing.
Reviewer 2 Report
This article reported the effectiveness and safety in the treatment of open-angle glaucoma (OAG) by using Hydrus Microstent and iStent Trabecular Bypass in combination with cataract phacoemulsification. As a prospective non-randomized comparative study, it convincingly showed that in the 24-month observation that both implants are safe ways in OAG patients.
Major concerns related to this paper:
1. As a prospective study, it would be better if the authors provide the registration number of this clinical trial.
2. The criteria of satisfactory success were uncommon, please describe the reasons.
3. In statistical methods, using the Student's t-test to verify the significance of differences between the means may increase the incidence of type â… errors, the univariate repeated measure ANOVA might be more appropriate.
4. In table 1, the paper didn’t list the type of OAG, such as POAG or JOAG? the number of antiglaucoma medication before surgery also could be added to.
5. In line 172, the number “71.7” may be “71.7%”?
6. In table 3, the numbers of patients with two implants were reduced at 24 months after surgery. What’s the reason for the reduce in these cases?
7. The authors mentioned “Figure 4 shows hazard ratio estimates for Hydrus and iStent at 24 months postoperatively” in line 202, it may be used to described “Figure 5”?
8. In this paper, the description of BCVA should be converted from decimal visual acuity to logMAR values.
Reviewer 3 Report
Present study compared the IOP lowering effect of two modern Schlemm’s canal based MIGSs, providing direct evidence for the clinical practice. Some comments were as following.
1. Purpose of the study should be explained in the introduction.
2. The authors stated that present study is a prospective non-randomized comparative design. Were the patients in two groups enrolled during the same period? Who made the decisions which surgery should be delivered to the patient?
3. Did this study included normal tension glaucoma?
4. It is noted that only 25 cases in iStent and 22 cases in hydrus group finished the 24-month follow-up. How about the other cases (missed or did not reach the visit time)?
5. Page 1, line 41: “If blood pressure targets are not reached and/or disease progresses despite topical medications, surgical IOP reduction is the next step [3].” Should be “ if the target IOP can not be reached……”
6. In table 2, the 95%CI for M24 in iStent was (14.64-7.12). Should it be 17.12?
7. Cataract alone is reported to results in 2 to 3mmHg IOP reduction in POAG eyes. In direct comparison of Hydrus and iStent without cataract extraction, Hydrus lowered IOP by 3.1 mm Hg (95% CI, 2.0 to 4.2 mm Hg) more than iStent alone. [1] It is consistently perceived that postoperative IOP reduction was greater in eyes with higher baseline IOP. The IOP reduction for iStent combined with cataract was reported at 28%.[2] In present study, IOP reduction in both groups was quite small (only 0.1-1.4mmHg). statistical significant difference in IOP was only observed at 12 month in iStent group. This may be related to the low IOP at baseline (16.0+/-3.2 in iStent group and 16.3 2.2 in hydrus group).
[1] Bicket AK, Le JT, Azuara-Blanco A, Gazzard G, Wormald R, Bunce C, Hu K, Jayaram H, King A, Otárola F, Nikita E, Shah A, Stead R, Tóth M, Li T.Minimally Invasive Glaucoma Surgical Techniques for Open-Angle Glaucoma: An Overview of Cochrane Systematic Reviews and Network Meta-analysis. JAMA Ophthalmol. 2021 Sep 1;139(9):983-989. doi: 10.1001/jamaophthalmol.2021.2351
[2] Paletta Guedes RA, Gravina DM, Paletta Guedes VM, Chaoubah A.Two-Year Comparative Outcomes of First- and Second-Generation Trabecular Micro-Bypass Stents with Cataract Surgery. Clin Ophthalmol. 2021 May 5;15:1861-1873. doi: 10.2147/OPTH.S302684. eCollection 2021.PMID: 33981138 .
8. Present study only provide the mean change in the number of anti-glaucoma medication after the surgery. Please provide the number of medications at baseline and after the surgery at different follow-up visit.
9. The results from present study showed that age of the patient below 70 years of age may have a more favorable effect on risk reduction in the Hydrus group than in the iStent group (HR = 0.81), while age above 70 years of age may reduce the risk in the iStent group (HR = 1.33). Mean age of present study (72.5 and73.5) seems quite older than other MIGSs studies. Please provide the number or percentage of patients of 70 years and older in each group.
10. Please pay more attention to the details in the manuscript (grammar, coma…).
